# Alleviation of Malathion Toxicity Effect by *Coffea arabica* L. Oil and *Olea europaea* L. Oil on Lipid Profile: Physiological and In Silico Study

**DOI:** 10.3390/plants10112314

**Published:** 2021-10-27

**Authors:** Khalid M. Al-Asmari, Isam M. Abu Zeid, Hisham N. Altayb, Atef M. Al-Attar, Mohammed Y. Alomar

**Affiliations:** 1Department of Biological Sciences, Faculty of Science, King Abdulaziz University, P.O. Box 80203, Jeddah 21589, Saudi Arabia; abuzeidmm@yahoo.com (I.M.A.Z.); atef_a_2000@yahoo.com (A.M.A.-A.); mohdaalomar@gmail.com (M.Y.A.); 2Department of Biochemistry, Faculty of Science, King Abdulaziz University, Building A 90, Jeddah 21589, Saudi Arabia; hdemmahom@kau.edu.sa; 3Princess Dr. Najla Bint Saud Al-Saud Center for Excellence Research in Biotechnology, King Abdulaziz University, P.O. Box 80200, Jeddah 21589, Saudi Arabia

**Keywords:** malathion, *Coffea arabica* oil, *Olea europaea* oil, lipid profile, chlorogenic acid, oleuropein

## Abstract

The community health plans commonly use malathion (MAL), an organophosphate pesticide (OP), to eliminate pathogenic insects. The objective of the present research is to evaluate the consequences of *Coffea arabica* L. oil and *Olea europaea* L. oil on MAL-intoxicated male rats. Six equal groups of animals were used for conducting this study (*n* = 10). Animals in group one were designated as control, animals belonging to group two were exposed to MAL in the measure of hundred mg per kg BW (body weight) for forty-nine days (seven weeks), rats in the third and fourth groups were administered with 400 mg/kg BW of *Coffea arabica* L. and *Olea europaea* L. oils, respectively, and the same amount of MAL as given to the second group. Groups five and six were administered with the same amount of *Coffea arabica* L. oil and *Olea europaea* L. oil as given to group three. Exposure of rats to 100 mg/kg body weight of MAL resulted in statistical alteration of the serum lipid profile. A marked decline was noticed in the severe changes of these blood parameters when MAL-intoxicated rats were treated with *Coffea arabica* L. oil and *Olea europaea* L. oil. Two compounds from *Coffea arabica* L. oil (Chlorogenic acid) and *Olea europaea* L. oil (Oleuropein) demonstrated good interaction with xanthine oxidase (XO) and 3-hydroxy-3-methyl-glutaryl-CoA reductase (HMGR) enzymes that are associated with cholesterol production. The present study indicated that *Coffea arabica* L. oil and *Olea europaea* L. oil could be considered prospective and potential healing agents against metabolic conditions induced by MAL.

## 1. Introduction

In the present time, the chemicals that are considerably used all over the world are pesticides. It is also one of the most toxic materials for humans. As the nature of pesticides is poisonous, it acts as a potential hazard for animals, human beings, various other organisms, and the environment. Even though pesticides are manufactured to kill only the intended insect, it has an adverse effect on the environment and human if not used correctly. The toxicity of pesticides can occur by unintentional ingestion, breathing in of its smoke, skin contact, and accidental eye exposure [1,2,3,4]. The organic compound malathion (MAL) containing phosphorus is a pesticide that is mostly employed for protecting field crops and has various other domestic applications. The ill-effect of malathion both on the environment and humans is a matter of serious concern [5]. The action of acetylcholinesterase gets inhibited by malathion [6]. The respiratory complex of the mitochondria gets inactivated because of the oxidative stress induced by malathion [7,8,9]. Many previous studies also mentioned that the exposure of malathion results in accelerated oxidative degradation of lipids in the RBCs (erythrocytes) of rats and their brain and liver [6,10,11].

The world is witnessing more and more cardiovascular disease (CVD). For the increased occurrence of CVD, it is regarded that the augmentation of LDL-c and total cholesterol are responsible [12]. Many reports have pointed out the changes in serum lipid levels occurring due to the toxicity of single pesticides [13,14,15]. However, very little information is available about the impact of the combination of pesticides on serum lipid levels. The pathophysiology of many diseases lays the blame on oxidative stress and the surge in lipid peroxidation (LPO). Therefore, the limelight has now been shifted to antioxidants, which demonstrate the scavenging action against reactive oxygen species (ROS) and restrain LPO. Antioxidants are employed as an important measure of action for preventing many diseases [16], together with pesticide toxicity [17].

The bioactive components of medicinal and edible plants are gaining much attention as a preventive measure and a remedy for or management of chronic ailments as they are either non-toxic or have very low toxicity [18,19,20]. Coffee, both as a cold and hot beverage, is most widely devoured all over the world. The epidemiological research conducted in 1960 and 1970 demonstrated that consuming more than or equal to three cups of coffee every day may help reduce various causative factors of overweight and obesity. Other phenolic compounds present in coffee (apart from caffeine) contribute to the effects like these [21]. Unroasted green coffee is rich in phenolic compounds. Consumption of green coffee is also linked with decreased risk of those diseases in which oxidative damage occurs [22]. On the other hand, *Olea europaea* L. oil extracted from olive fruit consists of both polyphenols and monounsaturated fatty acids. Even today, the health benefits of consuming *Olea europaea* L. oil remain unexcelled and are most likely attributed to the extraordinary chemical composition of *Olea europaea* L. oil [23]. *Coffea arabica* L. oil and *Olea europaea* L. oil contain a lot of bioactive compounds, including chlorogenic acid, oleic acid, linoleic acid, oleuropein, and hydroxytyrosol [24,25,26,27,28,29,30,31,32,33,34].

Stroke and heart ailments occur mainly because of hypercholesterolemia all over the world. The biosynthesis of uric acid utilizes a rate-limiting enzyme called xanthine oxidase, which is the root cause of generating ROS (reactive oxygen species), which initiates the formation of crystals of cholesterol and atherosclerosis. Another enzyme called 3-hydroxy-3-methyl-glutaryl-CoA reductase slows down the rate of reaction in cholesterol production. Though a potential reduction was observed in the cholesterol level by some enzyme-inhibiting chemicals that are easily available in the market, many of them do not meet the potential drug requirement for an individual person [35,36,37]. This study aims to evaluate the biochemical perturbations stimulated by malathion (MAL) in the serum lipids profile of rats and the protective influences of *Coffea arabica* L. oil and *Olea europaea* L. oil against the oxidative damage and stress caused by MAL intoxication.

## 2. Results

### 2.1. Assessment of Biochemical Markers

The results of this study demonstrated that the levels of TG (mmol/L) considerably (*p* ≤ 0.001) surged in the MAL-intoxicated G2 group of rats, as measured against the G1 (normal control) rats. Likewise, when measured against the G1 rats, the serum TG levels of MAL-poisoned rats ingested with the oil of *Coffea arabica* L. (G3) in the measures of 400 mg per kg BW, increased considerably (*p* ≤ 0.05). Conversely, a substantial reduction (*p* ≤ 0.001) was noticed in serum TG levels of the rats ingested with *Coffea arabica* L. (G3) and *Olea europaea* L. (G4), as compared to G2 rats (MAL-poisoned). While comparing it with the G1 group (normal control), the serum TG levels showed a non-significant variation in MAL-poisoned rats fed with *Olea europaea* L. oil (G4), as shown in Figure 1.

The serum cholesterol (mmol/L) levels considerably (*p* ≤ 0.001) increased as a result of MAL intoxication in the G2 group when a comparison was made with the G1 group of rats (normal control). Furthermore, while comparing it with the G1 group (normal control G1), the serum cholesterol levels considerably (*p* ≤ 0.001) increased when the MAL-poisoned rats were daily fed with the oils of *Coffea arabica* L. (G3) and *Olea europaea* L. (G4) in the measures of 400 mg/kg BW, alternatively, as compared to the MAL-intoxicated G2 rats, the serum levels of cholesterol significantly decreased in MAL-poisoned rats when fed with the oils of *Coffea arabica* L. (G3: *p* ≤ 0.01) and *Olea europaea* L. (G4: *p* ≤ 0.001), as shown in Figure 2.

A comparative analysis of the LDL-c (mmol/L) levels in the blood serum of every group (G1–G6) is presented in Figure 3. The serum LDL-c levels remarkably (*p* ≤ 0.001) increased in the G2 rats (MAL-intoxicated) when comparing it with the G1 group (normal control). As measured against the G1 group (normal control), the LDL-c levels showed a noticeable (*p* ≤ 0.05) surge in MAL-poisoned rats ingested with the oil of *Coffea arabica* L. (G3) daily when administered in the measures of 400 mg per kg body weight. However, a gavage of *Olea europaea* L. oil (G4) in the measures of 400 mg/kg BW to the MAL-poisoned rats demonstrated no considerable alterations in the serum, as measured against the G1 group (normal control). It was found that treating MAL-poisoned rats with the oils of *Coffea arabica* L. (G3) and *Olea europaea* L. (G4) in the measures of 400 mg/kg BW daily caused a considerable drop (*p* ≤ 0.01 and *p* ≤ 0.001, respectively) in LDL-c levels, as measured against the G2 group of rats (MAL-intoxicated, but not treated).

A substantial (*p* ≤ 0.001) increment was reported in serum VLDL-c (mmol/L) levels in G2 animals (MAL-intoxicated), as measured against the G1 group (i.e., the normal control). Compared with the G1 group (normal control), the ingestion of oils of *Coffea arabica* L. daily in the measures of 400 mg/kg BW to the rats poisoned with MAL (G3) caused a considerable increment (*p* ≤ 0. 05) in serum VLDL-c levels. In addition, when MAL-poisoned animals were administered with *Olea europaea* L. oil (G4) at a gavage of 400 mg per kg BW, it resulted in a non-significant variation VLDL-c levels, as measured against the normal control G1 group. This study confirmed that treating G2 (MAL-poisoned) rats with the oils of *Coffea arabica* L. (G3) and *Olea europaea* L. (G4) in the measures of 400 mg/kg BW caused a considerable (*p* ≤ 0.001) reduction in the serum VLDL-c levels, when compared with the G2 group, i.e., rats poisoned with MAL (Figure 4).

The results in Figure 5 indicated a marked reduction (*p* ≤ 0.001) in serum HDL-c (mmol/L) levels in G2 rats (MAL intoxicated) when measured against the G1 group (normal control). Likewise, the oral feeding of *Coffea arabica* L. oil (G3) and *Olea europaea* L. oil (G4) measuring at 400 mg per kg BW to the rats poisoned with MAL indicated a considerable reduction (*p* ≤ 0.01) in the HDL-c serum levels, as measured against G1 group. Conversely, treating MAL-intoxicated rats every day with *Coffea arabica* L. oil (G3) and *Olea europaea* L. oil (G4) measuring at 400 mg per kg BW demonstrated a considerable surge (*p* ≤ 0.01) in the HDL-c serum levels, when a comparison was made with the G2 group (MAL-intoxicated) of rats.

### 2.2. Molecular Docking Study

The docking of *Coffea arabica* L. oil compounds on XO and HMGR proteins showed good binding energy ranging from −10.6 to −3.1 kcal/mol. The redocking of the attached ligand (TEI-6720) showed a −10.8 kcal/mol docking score. Chlorogenic acid demonstrated the best binding energy with both XO and HMGR proteins with docking scores −8.2 and −10.6 kcal/mol, respectively. After chlorogenic acid, oleic acid showed the best binding energy with a −7 kcal/mol docking score for both XO and HMGR (Table 1). The Oleuropein from *Olea europaea* L. oil showed good binding energy (−11.36 kcal/mol), and three hydrogen bonds were formed with Asn650, Leu684, and Thr1010. Moreover, two bi-bi stacking bonds were formed with XO residues Phe914 and Phe 1009 (Figure 6A,B). The chlorogenic acid from *Coffea arabica* L. oil compounds and XO protein interaction resulted in a −8.2 kcal/mol docking score. This interaction resulted from one hydrogen bond with Thr1010 and two bi-bi stacking with Phe1009 and Phe914 (Figure 6C,D).

The HMGR (1HWK) protein showed the best interaction with chlorogenic acid (−10.6 kcal/mol) and oleuropein (−8 kcal/mol) (Table 1). There were eight hydrogen bonds formed with Ser565, Glu559, Arg590, lys691, Lys692, Ser684, Lys735, and Asn755 (Figure 7A,B). The oleuropein compound exhibited a −8 kcal/mol docking score and six hydrogen bonds with Lys691, Glu559 (three bonds), Ser684, Ser565 residues of HMGR protein (Figure 7C,D). The redocking of co-crystalized ligand (Atorvastatin) of HMGR protein showed a −12.3 kcal/mol docking score.

## 3. Discussion

In agricultural practices, pesticides are used to destroy pests and control the transmission of diseases, thereby enhancing food production. The community health plans commonly use malathion [O, O-dimethyl-S-(1,2-dicarcethoxyethyl) phosphorodithioate] which is an organophosphate pesticide (OP), to eliminate pathogenic pests [38,39]. Many previous research has indicated the toxic effect of malathion on both animals as well as human beings. It was also found in many previous studies that organophosphate pesticides can cause the production of ROS by damaging different cellular structures. The oxidative stress and the LPO associated with the toxicity caused by organophosphate pesticides can be reduced by treating it with minerals and antioxidants [40,41,42,43].

After studying human history, it was found that medicinal and herbal plants were basically used for treating various diseases. These traditional medications are still being employed by a wide range of populations for treating many health-related issues by various healthcare groups [44]. In developing nations, *Coffea arabica* L. is considered as one of the highest-producing tropical yields. Different chemicals having various benefits for overall health are found in the compounds of coffee. It is verified that different parts of the plant, including its oil, were used during ancient times for producing medicines to cure various diseases [45]. The tree of *Olea europaea* L. also holds a significant place in the history of medicines that have been in traditional use as a treatment for several diseases. The proof of this is the olive-rich diet of Mediterranean countries, which demonstrates the benefits of *Olea europaea* L. for reducing the onset of CVD and cancer [46]. The benefits of consuming Mediterranean foods on the general health of human beings, particularly for decreasing the risk of heart diseases, are mainly due to a high amount of extra virgin olives in these diets [47].

This study is a pioneer in investigating the efficacy and therapeutic effect of *Coffea arabica* L. oil and *Olea europaea* L. oil on malathion poisoning in male albino rats. This investigation found that the MAL exposure reduced the serum level of HDL-c, while the MAL-treated animals had significantly higher serum TG, cholesterol, VLDL-c, and LDL-c. These results confirm the previous research work, which demonstrated the severe disturbances in the metabolism of proteins, carbohydrates, and lipids when exposed to pesticides, including MAL [48,49,50,51,52,53,54,55].

The stimulated catecholamine triggers the lipolysis, which in turn elevates the production of fatty acids resulting in hyperlipidemia, as observed in this study. The present study also reports the increment in total serum levels of TG and cholesterol, which the blocking of bile ducts could cause. This results in a stoppage or reduction of its discharge into the intestine, thus causing cholestasis. Moreover, this could also be caused by the increased synthesis or decreased disintegration of lipids in the liver due to the disrupted activity of lipoprotein lipase [56,57]. Furthermore, the increase of serum LDL-c, VLDL-c, and decrease of high-density lipoprotein-c (HDL-c) levels in male albino rats is caused by the metabolic disorders occurring because of the destruction and necrosis of the hepatic cells [48,53,58,59].

The use of pesticides results in various adverse health effects. One of them is oxidative stress, which is linked with the toxicity mechanism in both animals and human beings. Researchers use this parameter as an important tool to monitor certain changes in their studies. Oxidative stress disrupts the stability of antioxidant resistance and ROS, which causes various changes in the LPO levels and antioxidant enzymes. Therefore, it can be said that reactive oxygen species have a major role in causing pesticide toxicity [60]. This investigation also established that feeding MAL-intoxicated animals with *Coffea arabica* L. or *Olea europaea* L. oil diminished the adverse effect of MAL on the biochemical parameters and maintained the healthy levels of lipid profiles like cholesterol, triglycerides, LDL-c, HDL-c, and VLDL-c. Moreover, it is observed that the tested medicinal plants can slow down the process of free radical generation and increase the action of antioxidants, thereby improving the oxidative damage caused by malathion.

The protective effects of *Coffea arabica* L. oil against several chemical toxicants linked with the disruption of oxidative damage caused in investigated animals were reported by earlier research studies [61,62,63]. Green coffee causes a clear reduction of cholesterol, triglycerides, VLDL-c, LDL-c and elevates the production of HDL-c [64]. Previous studies have concluded that *Olea europaea* L. oil reduces serum cholesterol, TG, and low-density lipoprotein cholesterol levels and enhances the production of HDL-c in rats [65,66], mice [67], and guinea pigs [68]. Consuming the oil of *Olea europaea* L. is considered beneficial for health because it is loaded with abundant MUFA (monounsaturated fatty acids) and phenolic compounds, both of which are regarded healthy [69,70,71]. Both phenolic compounds and MUFA demonstrate lipid-lowering action, thereby helping prevent the oxidation of low-density lipoprotein cholesterol. Therefore, *Olea europaea* L. oil (rich in MUFA and phenolic compounds) is said to adjust the biochemical parameters of lipid within the healthy range. Mediterranean foods containing the oil of *Olea europaea* L. diminish the chances of getting heart diseases [72,73,74,75]. However, the protection offered by *Coffea arabica* L. oil or *Olea europaea* L. oil against the damage caused by ROS should also be included. They enhance the defense mechanism of antioxidants and reduce the severe alterations that occurred in biochemical and hematological parameters of MAL-poisoned male rats.

Two compounds from *Olea europaea* L. oil (oleuropein) and *Coffea arabica* L. oil (chlorogenic acid) demonstrated good interaction with XO and HMGR enzymes that are associated with cholesterol production [26]. The oleuropein showed a better docking score from the known inhibitor (TEI-6720) of XO protein, and chlorogenic acid showed a redocking score near to the known anti-cholesterol drug Atorvastatin. Other studies, which showed the anti- hypercholesterolemic effect of chlorogenic acid [76,77,78], support the present research findings. In conclusion, this study demonstrated the potential of two natural products, i.e., *Coffea arabica* L. oil and *Olea europaea* L. oil, for treating malathion poisoning as they are antioxidants in nature.

## 4. Materials and Methods

### 4.1. Plant Oils

*Coffea arabica* oil was obtained from Fushi, London, United Kingdom, and *Olea europaea* oil was obtained from the Aljouf region, Saudi Arabia.

### 4.2. Animals Model

This experiment was conducted on albino male rats belonging to the strain Wistar (Rattus norvegicus). The rats were 110–144 g in weight. The Mansour Scientific Research and Development Foundation (MSRDF), located in Jeddah, Saudi Arabia, provided the animals for conducting the experiment. Before starting the experiment, the animals were made accustomed to the laboratory environment for about seven days (one week). Standard cages made up of plastic were used for keeping the experimental rats. Additionally, the controlled environment of the laboratory was retained at 20 ± 1 °C, 65% humidity, and a dark-light cycle of 12:12 h. Commercially available normal chow was given to the rats’ ad libitum (as desired) along with the 24 × 7 water availability. The guidelines of the animal ethics released by the Animal Care and Use Committee (ACUC) of King Abdulaziz University were followed for conducting the experiments. Moreover, all experiments were conducted in compliance with the Arrive guidelines and in accordance with the EU Directive 2010/63/EU for animal experiments.

### 4.3. Experimental Protocol

A random division of sixty albino rats was done in six equal groups (ten animals in each group). The following treatment was given to each group:No treatment was given to the rats in the first group (group 1). They were designated as uninterfered normal control.Oral administration of MAL (100 mg per kg BW) for 49 days (7 weeks) was provided to the rats of the second group (group 2).Oral supplementation of *Coffea arabica* L. oil (400 mg per kg BW) and exposure to MAL (100 mg per kg BW) after three hours of *Coffea arabica* L. oil ingestion was daily provided to the animals of the third group (group 3) for 49 days (7 weeks).Oral supplementation of *Olea europaea* L. oil (400 mg per kg BW) and exposure to MAL (100 mg per kg BW) after three hours of *Olea europaea* L. oil ingestion was daily provided to the animals of the fourth group (group 4) for 49 days (7 weeks).Oral supplementation of *Coffea arabica* L. oil (400 mg per kg BW) was provided daily to the animals of the fifth group (group 5) for 49 days (7 weeks).Oral supplementation of *Olea europaea* L. oil (400 mg per kg BW) was provided daily to the animals of the sixth group (group 6) for 49 days (7 weeks).

### 4.4. Analysis of Blood Serum

The rats under study were subjected to a twelve-hour fasting period with free access to water after 49 days (7 weeks). Then, diethyl ether was given to them as an anesthetic drug. The venous plexus in the orbital region was used to collect the blood sample, which was then placed without heparin in centrifuge tubes. The tubes were then subjected to centrifugation at 2500 rpm for 15 min. After the centrifugation process, the serum collection was done from the centrifuge tubes and put in storage at −80 °C. The levels of triglycerides (TG), total cholesterol, LDL-c, and HDL-c were estimated using the methodologies proposed by [79,80,81,82], respectively. The equation given was used to evaluate (VLDL-c): VLDL-c = TG/2.175.

### 4.5. BIOINFORMATICS Analysis

#### 4.5.1. Selection of Enzymes for Docking Study

3-hydroxy-3-methyl-glutaryl-CoA reductase (HMGR) was chosen for docking studies because of its involvement in catalyzing the conversion process of HMG-Co-A to mevalonate, hence resulting in a reaction that limits the rate of cholesterol production. The xanthine oxidase enzyme is hypothesized to be the source of ROS that causes cholesterol formation [26].

#### 4.5.2. Molecular Docking Study

Protein crystal structures were obtained from the PDB database, with the following accession numbers: PDB ID: 1HWK for HMGR and PDB ID: 1N5X for XO. The top compounds found in *Coffea arabica* L. oil and *Olea europaea* L. oil [24,25,26,27,28,29,30,31,32,33,34] were obtained from PubChem database as shown in Table 1.

Schrodinger Suite 2021-3, containing the Maestro program, was used for molecular docking study. Additionally, LigPrep, Receptor grid generation, and SiteMap interfaces of the Maestro program were used for ligands preparation, proteins preparation, and active site prediction (Schrodinger, LLC, New York, NY, USA, 2021). Proteins were set as rigid, while ligands were set as flexible. The attached ligands (Atorvastatin and TEI-6720) with HMGR and XO receptors were then redocked again for docking protocol validity.

### 4.6. Analysis of the Statistics

The software IBM SPSS version 24 of the Statistical Package for Social Sciences was used for statistical analysis. The results show the means ± standard error (SE). The general core of the work is to compare between the groups, either between all groups or between groups 2, 3, and 4. The statistical technique called one-way ANOVA was employed for comparing the six groups. LSD (least significant difference) was then used as a post-hock test when ANOVA showed the significant difference between groups. When *p*-value ≤ 0.05, it was regarded as a statistically considerable difference.

## 5. Conclusions

This research demonstrated the potential of two natural products, i.e., *Coffea arabica* L. oil and *Olea europaea* L. oil, for treating malathion poisoning as they are antioxidants in nature. Generally, rats administered with *Olea europaea* L. oil showed the maximum protective action followed by *Coffea arabica* L. oil. Nevertheless, intense study on various biochemical and hematological parameters is still required for investigating and establishing the effectiveness of altered dosage and concentration of both *Coffea arabica* L. oil and *Olea europaea* L. oil against MAL toxicity. Moreover, computational analysis shows cessation of HMGR and XO by *Coffea arabica* L. oil (chlorogenic acid) and *Olea europaea* L. oil (oleuropein). More in vivo studies are required for the validation of our findings. The alternative ways for treating hypercholesterolemia are still needed, which is why the exploration of clinical efficiency of secondary molecules through further research and thorough study has become a necessity.

## Figures and Tables

**Figure 1 plants-10-02314-f001:**
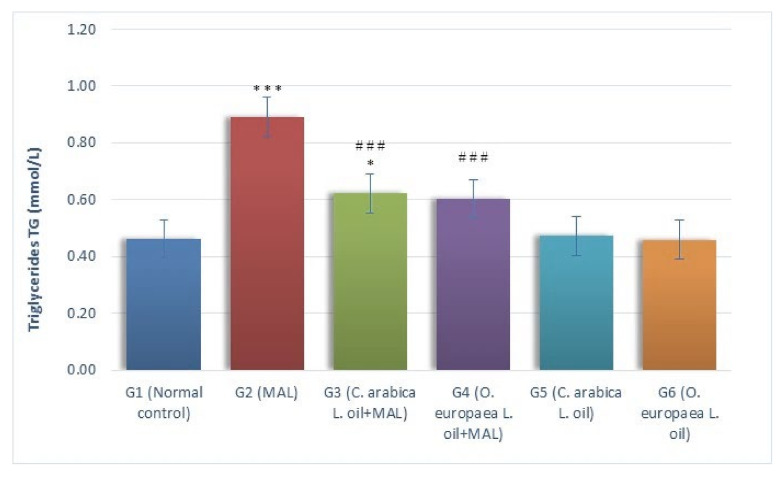
Influence of *Coffea arabica* L. oil and *Olea europaea* L. oil on the serum triglyceride levels. When a comparison was made against the normal control group, a significant difference was found in the mean values at *p* ≤ 0.05 *; *p* ≤ 0.001 ***. When a comparison was made against the MAL-intoxicated rats, a significant difference was found in the mean values at *p* ≤ 0.001 ^###^.

**Figure 2 plants-10-02314-f002:**
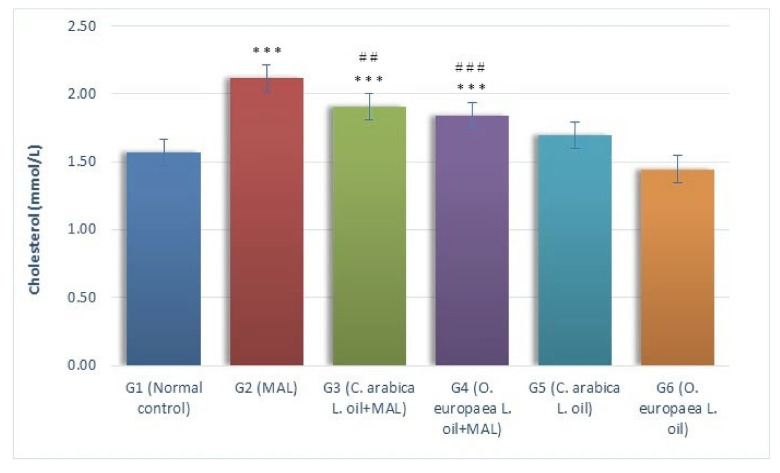
Influence of *Coffea arabica* L. oil and *Olea europaea* L. oil on the level of cholesterol. When a comparison was made against the normal control group, a significant difference was found in the mean values at *p* ≤ 0.001 ***. When a comparison was made against the MAL-intoxicated group, a significant difference was found in the mean values at *p* ≤ 0.01 ^##^; *p* ≤ 0.001 ^###^.

**Figure 3 plants-10-02314-f003:**
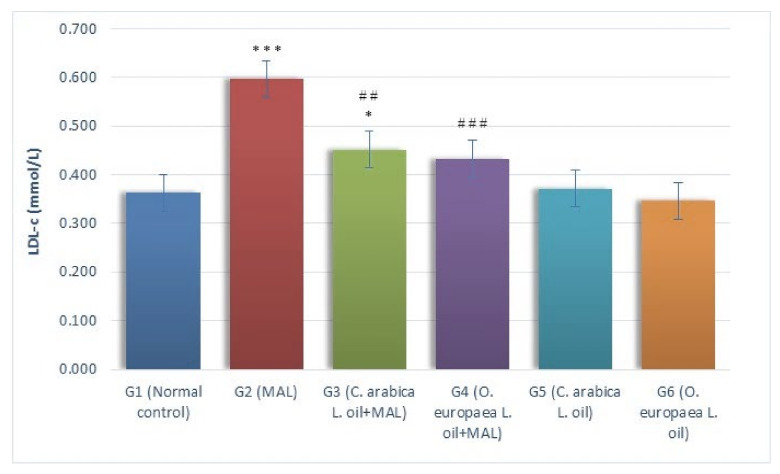
Influence of *Coffea arabica* L. oil and *Olea europaea* L. oil on the serum LDL-c levels. When a comparison was made against the normal control group, a significant difference was found in the mean values at *p* ≤ 0.05 *; *p* ≤ 0.001 ***. When a comparison was made against the MAL-intoxicated rats, a significant difference was found in the mean values at *p* ≤ 0.01 ^##^; *p* ≤ 0.001 ^###^.

**Figure 4 plants-10-02314-f004:**
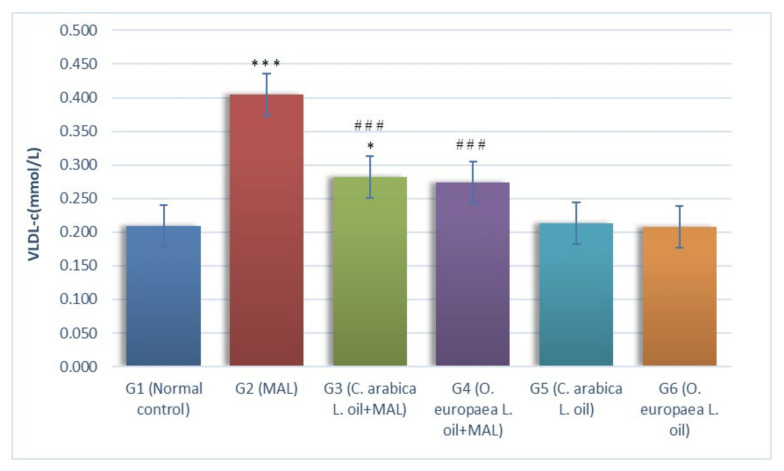
Influence of *Coffea arabica* L. oil and *Olea europaea* L. oil on the level VLDL-c. When a comparison was made against the normal control rats, a significant difference was found in the mean values at *p* ≤ 0.05 *; *p* ≤ 0.001 ***. When a comparison was made against the MAL-intoxicated rats, a significant difference was found in the mean values at *p* ≤ 0.001 ^###^.

**Figure 5 plants-10-02314-f005:**
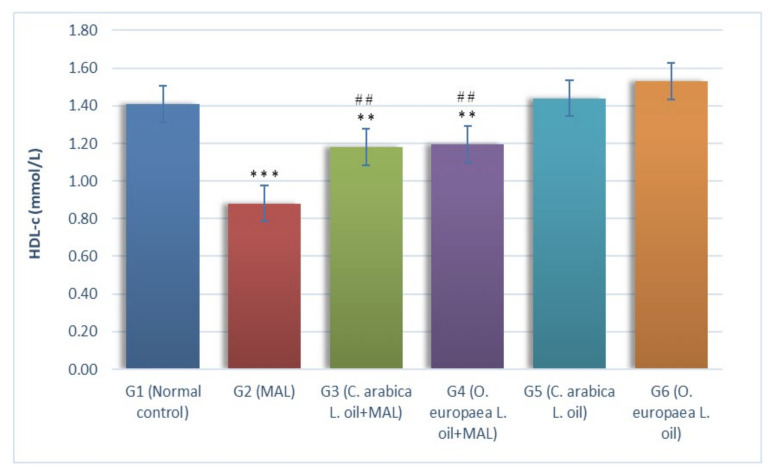
Influence of *Coffea arabica* L. oil and *Olea europaea* L. oil on the level HDL-c. When a comparison was made against the normal control group, a significant difference was found in the mean values at *p* ≤ 0.01 **; *p* ≤ 0.001 ***. When a comparison was made against the MAL-intoxicated rats, a significant difference was found in the mean values at *p* ≤ 0.01 ^##^.

**Figure 6 plants-10-02314-f006:**
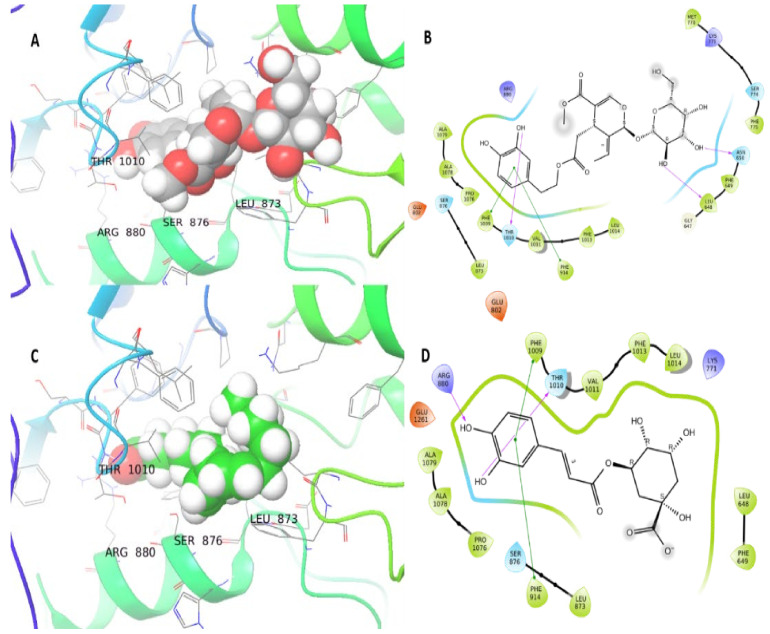
3D and 2D interaction of XO protein, the ligands are shown in balls at 3D representation, and the protein backbones are shown in ribbons, hydrogen bonds are shown in purple arrows, bi-bi stacking is shown in green lines. (**A**,**B**): Three-dimensional and two-dimensional interface of oleuropein. (**C**,**D**): Three-dimensional and two-dimensional interface of XO protein and chlorogenic acid ligand.

**Figure 7 plants-10-02314-f007:**
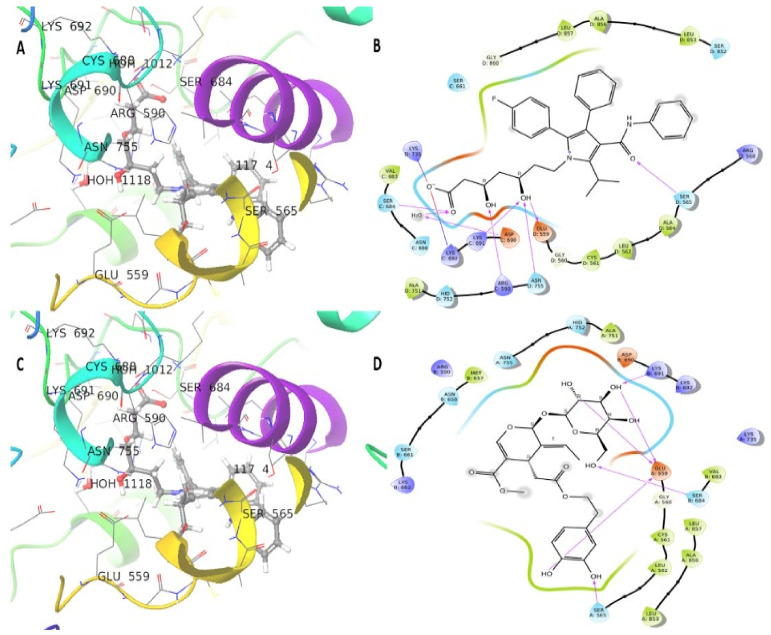
3D and 2D interaction of HMGR (1HWK) protein, the ligands are shown in black at protein center, and the protein backbones are shown in ribbons, hydrogen bonds are shown in purple arrows, bi-bi stacking is shown in green lines. (**A**,**B**): Three-dimensional and two-dimensional interface of chlorogenic acid. (**C**,**D**): Three-dimensional and two-dimensional interface of HMGR protein and oleuropein ligand.

**Table 1 plants-10-02314-t001:** The docking of *Coffea arabica* L. oil and *Olea europaea* L. oil compounds on XO and HMGR.

Compounds	PubChem ID	XO (1N5X)	HMGR (1HWK)
*Coffea arabica* L. oil compounds			
Chlorogenic acid	1794427	−8.2	−10.6
Oleic acid	445639	−7	−7
Linoleic acid	5280450	−6.7	−6.7
Palmitic acid	985	−6.2	−6.2
Kahweol	114778	−5.5	−3.8
Cafestol	108052	−5.2	−3.8
Caffeine	2519	−3.4	−3.1
*Olea europea* L. oil compound			
Oleuropein	5281544	−11.36	−8
Linoleic acid	5280450	−8.3	−6.7
Oleic acid	445639	7.3	−7
Palmitic acid	985	−6.2	−6.2
Stearic acid	5281	−6.7	−6
Hydroxytyrosol	82755	−5.3	−5.4
Tyrosol	10393	−5.12	−4.5
Squalene	638072	−5	−1.9
Controls			
Atorvastatin	60823		−12.3
TEI-6720	134018	−10.8	

## Data Availability

All relevant data are within the manuscript. All data were statistically analyzed, as mentioned in the submitted manuscript.

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
