# Peer review of "Alleviation of Malathion Toxicity Effect by *Coffea arabica* L. Oil and *Olea europaea* L. Oil on Lipid Profile: Physiological and In Silico Study"

_plants, 2021, doi:10.3390/plants10112314_

Round 1

Reviewer 1 Report

In the article "Alleviation of Malathion Toxicity Effect by Coffea arabica L. Oil and Olea europaea L. Oil on Lipid Profile: Physiological and in silico Study" the Authors present results of intriquing investigation wehere Coffea arabica L. oil and Olea europaea L. tested as protctive agent aginst malathion toxic effects. Presented problem and results are interesting and in intriguing. Article is well writen and organised. I have few considerations which I listed below:
  • in my opinnion content from disscussion should be moved to introduction. I think about explenation why this experiment was conducted (line 217-245). Interaction between malathion, plant oils and and lipids metabolism is not obvious that is why it should be precisely described in introduction. Some working theses, or questions could be useful too.
  • I think that there should be more concentrations od MAL and plant oils used to chcecked how animals react to treatment, what is the dynamics and direction of changes. With single concentration it is impossible
  • in figures standerd errors (SE) should be changed to standard deviations (SD)
  • I recommend to change colors in figures columns to more subdued
  • in materials and methods Authors should explain what was the source and quality of plant oils, and how food was prepared. There is only plant species described, but we know that place and conditions of growth can influence on compund profile. Exactly what varieties of plants were used.
  • I am no t convinsed about docking modeling. Authors did not make any analysis of chemical profile for plant oils. So how can be they sure that compounds used in modeling are present in oils? In this case, for proper conclusions any composition analysis should be made.
  • what was the level of food consumption? I something was added to food it can modulate taste/smell properties. How can we be sure that it not change and inluence on lipid levels in blood?
  • line 279 what exactly type of food was used? what was the composition (lipids, protein, carbohyrates)?
  • line 272 what Authors know about sexual identyty of rats? Gender refers to the characteristics of women, men, girls and boys that are socially constructed. Word "sex" should be used.
  • line 312 more information about used methods sholud be added . Just generaly, were they spectrophotometric, immunodetective chromatographic or other? What devices were used?
  • disscussion is finished wihout any closing/summary part. I recomend to conclude shortly whole project
  • I am not convinced to information about results of statistical analysis. With used decription there is no information if oil treated groups are different from MAL and MAL+oil treated group. There should be used multicomparison ANOVA test.

Author Response

In the article "Alleviation of Malathion Toxicity Effect by Coffea arabica L. Oil and Olea europaea L. Oil on Lipid Profile: Physiological and in silico Study" the Authors present results of intriguing investigation where Coffea arabica L. oil and Olea europaea L. tested as protective agent against malathion toxic effects. Presented problem and results are interesting and in intriguing. Article is well written and organized. I have few considerations which I listed below:

Point 1: In my opinion content from discussion should be moved to introduction. I think about explanation why this experiment was conducted (line 217-245). Interaction between malathion, plant oils and and lipids metabolism is not obvious that is why it should be precisely described in introduction. Some working theses, or questions could be useful too.

Response 1: Because some journals require a brief and precise introduction, line 217-245 discusses the relationship between malathion, plant oils, and lipid metabolism.

Point 2: I think that there should be more concentrations of MAL and plant oils used to checke how animals react to treatment, what is the dynamics and direction of changes. With single concentration it is impossible

Response 2:

We chose the MAL concentration (100 mg/kg) and tested plants (400 mg/kg) based on previous research.

We noticed that these amounts were effective in our preliminary experiments.

These concentrations were also chosen depending on the age and weight of rats available at the time of the experiment (COVID-19 time).

Furthermore, it was not our goal to compare the varied concentrations of MAL, olive and coffee oils.

Point 3: In figures standard errors (SE) should be changed to standard deviations (SD)

Response 3:

Both SD and SE are in the same units (the units of the data) and both are measures of variation. If we want to indicate the uncertainty around the estimate of the mean measurement, we quote the standard error of the mean. The standard error is most useful as a means of calculating a confidence interval (Altman and Bland, 2005).

Altman, D. G., & Bland, J. M. (2005). Statistics notes: standard deviations and standard errors. British Medical Journal331(7521), 903.

This is why we used SE and not SD.

Point 4: I recommend to change colors in figures columns to more subdued

Response 4: Done

Point 5: In materials and methods authors should explain what was the source and quality of plant oils, and how food was prepared. There is only plant species described, but we know that place and conditions of growth can influence on compund profile. Exactly what varieties of plants were used.

Response 5: Done

Point 6: I am not convinced about docking modeling. Authors did not make any analysis of chemical profile for plant oils. So how can be they sure that compounds used in modeling are present in oils? In this case, for proper conclusions any composition analysis should be made.

Response 6: From previous data the chemical profile of tested plant oils was extensively studies as shown in our article and most of them are normally available in plants. We selected the top chemicals for docking study. Moreover, in response to reviewer and for validation of our findings the following paragraph has been added to the conclusion section "more in vivo studies are required for validation of our findings."

Point 7: What was the level of food consumption? I something was added to food it can modulate taste/smell properties. How can we be sure that it not change and influence on lipid levels in blood?

Response 7: The food given to test animals was standard diet used in animal experiments ad libitum (as desired) with the 24x7 water availability. In our preliminary experiments there is no change and influence on lipid levels in blood.

Point 8: Line 279 what exactly type of food was used? What was the composition (lipids, protein, carbohydrates)?

Response 8: Animal feed for laboratory animals No. 1005 (Net Weight 50 kg) from Saudi grains organization (SAGO):

The components of diet:

Raw protein 20%, raw fat 4%, raw fibers 3.50%, ash 6%, salt 0.5%, Calcium 1%, phosphorous 0.6%, Vitamin A 20 IU/gm, Vitamin D 2.20 IU/gm, Vitamin E 70 IU/kg and energy 2850 Kilocalories/Kg.

Point 9: Line 272 what Authors know about sexual identity of rats? Gender refers to the characteristics of women, men, girls and boys that are socially constructed. Word "sex" should be used.

Response 9: Done

Point 10: Line 312 more information about used methods sholud be added. Just generaly, were they spectrophotometric, immunodetective chromatographic or other? What devices were used?

Response 10: The serum concentration of TG, cholesterol, HDL, LDL, and VLDL were determined using sandwich enzyme-linked immunosorbent assay (ELISA) rat specific kits of My BioSource, USA. Briefly, 96-well plates coated with specific antibodies against TG, cholesterol, HDL, LDL, and VLDL which will bind to the TG, cholesterol, HDL, LDL, and VLDL present in the serum and this is detected by a biotin conjugated antibody. The biotin moiety is subsequently detected, by the addition of streptavidin coupled horseradish peroxidase (HRP). ELISA detection substrate (3, 3’, 5, 5’-Tetramethylbenzidine, TMB) was used to visualize HRP enzymatic reaction. TMB was catalyzed by HRP to produce a blue colour product that changed into yellow after adding acidic stop solution. The density of yellow colour is directly proportional to the concentration of TG, cholesterol, HDL, LDL, and VLDL.

Point 11: Discussion is finished without any closing/summary part. I recommend to conclude shortly whole project.

Response 11: Done

Point 12: I am not convinced to information about results of statistical analysis. With used decription there is no information if oil treated groups are different from MAL and MAL+oil treated group. There should be used multicomparison ANOVA test.

Response 12: Our research goals are as follows:

  • To study the protective effects of tested oils (Olive and Coffee oils) against MAL toxicity by comparing Group 1 (Normal control group) to MAL+oil Groups (3 and 4).
  • To prove that the oils are safe and have no side effects by comparing  Group 1 (Normal control) with  Groups 5 and 6 (Rat oil).
  • This is why we used this type of statistical analysis.

Reviewer 2 Report

The article described the effects of Coffea arabica L. oil and Olea europaea L. oil. on MAL-intoxicated male rats and silico molecular docking. This is a timely and interesting paper.

However, I would like to mention some things:

There is a need to improve the flow of introduction. For example, in the third paragraph, the subject has been changed abruptly from cardiovascular disease to coffee as a beverage.

Line 67-68: “consuming Olea europaea L. oil in place of fats of other origins” -  too general statement

The introduction does not refer to docking study

Top compounds concentrations should be presented

The authors conducted the molecular docking study based on several predominant compounds. How about other phytochemicals that were not incorporated into the method?

The docking of main ingredients was meaningless. It can’t be proved that main ingredients are related with inhibitory activities without conducting experiments on inhibitory activities of the main ingredients. Perhaps, the minor ingredients have significant inhibitory activities.

Author Response

The article described the effects of Coffea arabica L. oil and Olea europaea L. oil. on MAL-intoxicated male rats and silico molecular docking. This is a timely and interesting paper. However, I would like to mention some things:

Point 1: There is a need to improve the flow of introduction. For example, in the third paragraph, the subject has been changed abruptly from cardiovascular disease to coffee as a beverage.

Response 1: Done

In response to Reviewer 2 and to improve the flow of introduction “Cardiovascular disease was relocated to the first paragraph”

Point 2: Line 67-68: “consuming Olea europaea L. oil in place of fats of other origins” - too general statement.

Response 2: Done

We already stated that the exceptional chemical makeup of Olea europaea L. oil is responsible for its health advantages.

Point 3: The introduction does not refer to docking study.

Response 3: There is information regarding the proteins used in this work in the last paragraph of the introduction. Also we referred to docking study in the materials and methods, results and discussion.

Point 4: Top compounds concentrations should be presented.

Response 4: As we indicated in our article, in the tested plants there are many bioactive chemicals based on past findings. Furthermore, diverse concentrations are employed for these compounds, making it impossible to mention a single concentration. This is why we selected the seven top chemicals in Coffee oil and eight in olive oil.

Point 5: The authors conducted the molecular docking study based on several predominant compounds. How about other phytochemicals that were not incorporated into the method?

Response 5: Yes, we only examined the most common top chemicals identified in the tested plants, and we chose these based on previous research (Reference 24- Reference 34), as stated in our paper. Although a variety of additional chemicals were detected in modest amounts in the plants studied, their presence was not prevalent.

Point 6: The docking of main ingredients was meaningless. It can’t be proved that main ingredients are related with inhibitory activities without conducting experiments on inhibitory activities of the main ingredients. Perhaps, the minor ingredients have significant inhibitory activities.

Response 6: The computation research is simply referred to as predictive; nevertheless, additional in vivo investigations are required for validation of our findings; in response to reviewer: the following paragraph has been added to the conclusion section "more in vivo studies are required for validation of our findings." The previous comment indicated the choosing of the major compounds and not minor ingredients.

Round 2

Reviewer 2 Report

Accept in present form